# MITIGATING COMPOSITIONAL ISSUES IN TEXT-TO-IMAGE GENERATIVE MODELS VIA ENHANCED TEXT EMBEDDINGS

## ABSTRACT

Text-to-image diffusion-based generative models have the stunning ability to generate photo-realistic images and achieve state-of-the-art low FID scores on challenging image generation benchmarks. However, one of the primary failure modes of these text-to-image generative models is in composing attributes, objects, and their associated relationships accurately into an image. In our paper, we investigate this compositionality-based failure mode and highlight that imperfect text conditioning with CLIP text-encoder is one of the primary reasons behind the inability of these models to generate high-fidelity compositional scenes. In particular, we show that (i) there exists an optimal text-embedding space that can generate highly coherent compositional scenes showing that the output space of the CLIP text-encoder is sub-optimal, and (ii) the final token embeddings in CLIP are erroneous as they often include attention contributions from unrelated tokens in compositional prompts. Our main finding shows that the best compositional improvements can be achieved (without harming the model's FID score) by fine-tuning *only* a simple and parameter-efficient linear projection on CLIP's representation space in Stable-Diffusion variants using a small set of compositional image-text pairs. This result demonstrates that the sub-optimality of the CLIP's output space is a major error source. We also show that re-weighting the erroneous attention contributions in CLIP can lead to slightly improved compositional performances.

## 1 INTRODUCTION

Text-to-image diffusion-based generative models (Rombach et al., 2021; Podell et al., 2023; Ramesh et al., 2021; Saharia et al., 2022) have achieved photo-realistic image generation capabilities on user-defined text prompts. However recent works (Huang et al., 2023) have designed compositionality benchmarks to show that these text-to-image models have low fidelity to simple compositionality prompts such as those consisting of attributes, objects, and their associated relations (e.g., "*a red book and a yellow vase*"). This hinders the use of these generative models in various creative scenarios where the end-user wants to generate a scene where the composition is derived from words (and their relationships) in the prompt.

Existing works (Chefer et al., 2023a; Feng et al., 2023; Agarwal et al., 2023; Wang et al., 2023) propose various ways to improve compositionality in text-to-image models. These works primarily rely on modifying the cross-attention maps by leveraging bounding box annotations and performing a small optimization in the latent space during inference. Recent methods based on fine-tuning (Huang et al., 2023) the UNet also lead to improved compositonality. Despite the progress, the *core reasons* behind compositionality failures in text-to-image models remain unclear. Understanding these reasons helps designing effective methods that can augment text-to-image models with improved compositional capabilities.

In our paper, we investigate possible reasons behind compositionality failures in text-to-image generative models. We identify two sources of errors: (i) We observe that output token embeddings in CLIP have

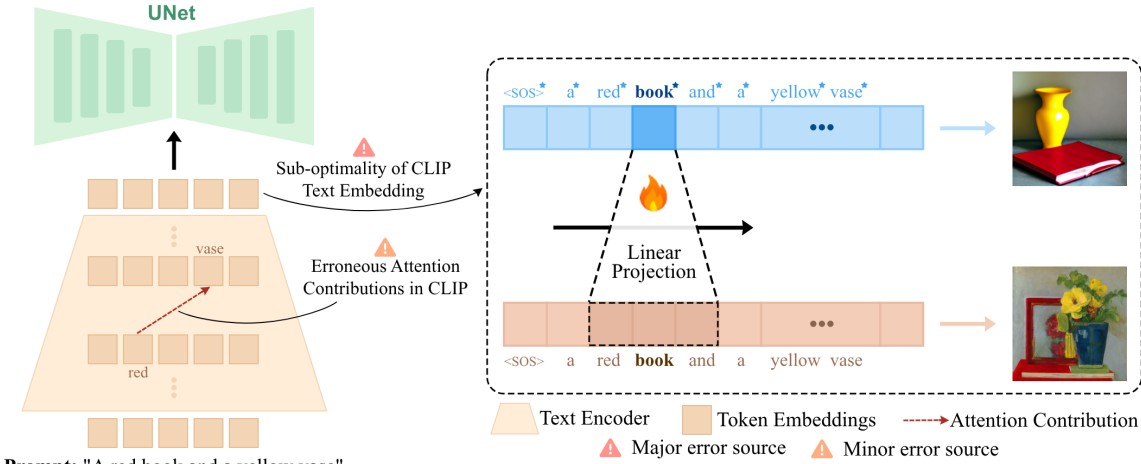

**Prompt:** "A red book and a yellow vase"

Figure 1: Overview of our analysis and proposed methods. The figure identifies two sources of errors in Stable Diffusion's inability to generate compositional prompts: (i) erroneous attention contribution in CLIP (minor) and (ii) sub-optimal CLIP text embedding (major). We propose a window-based linear projection (`WiCLP`), applying linear projection to a token's surrounding window to enhance embeddings.

significant attention contributions from irrelevant tokens, thereby introducing errors in generation. We then compare the internal attention contributions in CLIP for compositional prompts to the T5 text-encoder which has been shown to display strong compositional capabilities in DeepFloyd[1]. We quantitatively find that the T5 text-encoder displays significantly lesser erroneous attention contributions than CLIP, highlighting a potential reason towards its improved compositionality. (ii) Sub-optimality of CLIP output space on compositional prompts: We observe that optimizing the text embeddings, while utilizing a frozen Stable-Diffusion UNet, effectively generates images with compositional scenes. We find out that there exists a text-embedding space capable of generating highly coherent images with compositional scenes for various attributes (e.g., *color, texture, shape*) which highlights that the *existing CLIP output space is sub-optimal*. These results indicate that the output space of the CLIP text-encoder could be further improved to enable text-to-image models to generate more accurate compositional scenes.

Leveraging our observations on the deficiencies of the CLIP output space, we show that we can improve the output space of the CLIP text-encoder to better align with the optimal space by applying a *simple* linear projection on top of CLIP (see Figure 1). This leads to stronger compositional performances. In particular, we propose Window-based Compositional Linear Projection (`WiCLP`), a *lightweight* fine-tuning method that significantly improves the model's performance on compositional prompts, yielding results comparable to existing baselines (see Figure 2). Moreover, it preserves the model's clean accuracy, as evidenced by a low FID on clean prompts, offering a *parameter* and *speed*-efficient solution. We also show that reweighting the erroneous attention contributions in CLIP can lead to improved compositional performances, however, the improvements often lag behind `WiCLP`.

Fine-tuning a subset of components of the diffusion model can result in an increase in the FID score for clean prompts. While fine-tuning only a linear projection partially mitigates this, we find that applying it over all the time steps results in an increase in FID. To mitigate this, we introduce SWITCH-OFF where we only apply `WiCLP` during the initial steps of generation, switching it off for the remaining steps. This enables the model

[1]https://huggingface.co/DeepFloyd/IF-I-M-v1.0

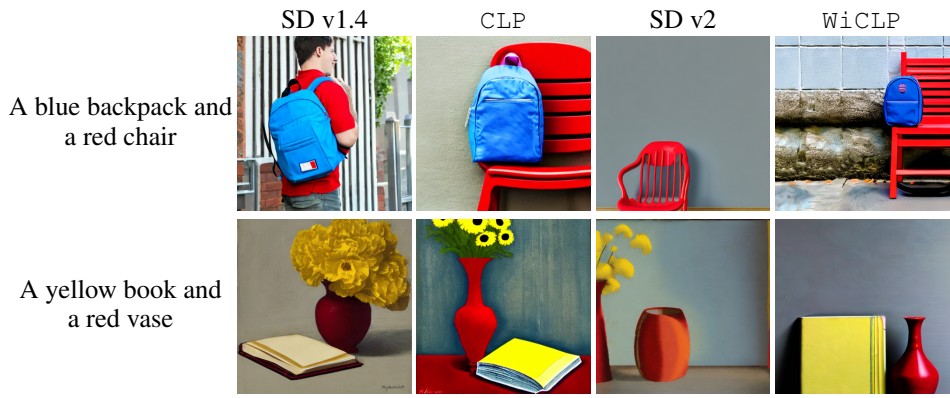

Figure 2: Qualitative comparison between `CLP` and `WiCLP` vs the baselines.

to obtain a coherent compositional scene in early steps (crucial for compositional prompts) while retaining clean accuracy on surrounding prompts, as the generation in final steps is guided by the original text-encoder not the augmented one that maps to the optimized space.

In summary, our contributions are as follows:

- We perform an in-depth analysis of the reasons behind compositionality failures in open-source text-to-image generative models, highlighting two reasons for them.
- Leveraging our observations, we propose `WiCLP` for Stable Diffusion v-1.4 and v-2 which can augment the models with improved compositionality while preserving their clean accuracy on surrounding prompts. We observe improvements of $16.18\%, 15.15\%$, and $9.51\%$ on SD v1.4 and $14.35\%, 11.14\%$, and $6\%$ on SD v2 in VQA scores (Huang et al., 2023) across color, texture, and shape datasets, respectively. Our method achieves competitive VQA scores compared to other baselines, while demonstrating superior FID on clean prompts, requiring *fewer parameters* for optimization, and enabling *fast inference*.

Overall, our paper provides quantitative evidence elucidating the compositional challenges within text-to-image models and strong baselines to mitigate such issues.

## 2 BACKGROUND

**Compositionality in Text-to-Image Generative Models.** A recent work Huang et al. (2023) introduces a benchmark for testing compositionality in text-to-image models showing the susceptibility of open-source text-to-image models on simple compositional prompts. In addition, the authors also propose a fine-tuning baseline to augment text-to-image models with improved compositionality. The compositionality issue can also be addressed at inference time by modifying the cross-attention maps leveraging hand-crafted loss functions and bounding boxes generated from a language model (Chefer et al., 2023a; Feng et al., 2023; Agarwal et al., 2023; Wang et al., 2023; Nie et al., 2024; Lian et al., 2023; Liu et al., 2022a). However, Huang et al. (2023) show that a data-driven and fine-tuning approach is more suitable towards improving compositionality.

**Interpretability of Text-to-Image Generative Models.** There have been recent efforts to interpret text-to-image models like Stable Diffusion. DAAM (Tang et al., 2023; Hertz et al., 2022) studies the generation

process in diffusion models by analyzing cross-attention maps between text tokens and image pixels, highlighting their semantic precision. Basu et al. (2023) use causal tracing to understand how knowledge is stored in models like Stable Diffusion v1 while Rezaei et al. (2024) propose a mechanistic approach to localize knowledge in cross-attention layers of various text-to-image models. Chefer et al. (2023b) explore concept decomposition in diffusion models.

## 2.1 TEXT-TO-IMAGE DIFFUSION MODELS: TRAINING AND INFERENCE

In diffusion models, noise is added to the data following a Markov chain across multiple time-steps $t \in [0, T]$. Starting from an initial random real image $x_0$ along with its caption $c$, $(x_0, c) \sim \mathcal{D}$, the noisy image at time-step $t$ is defined as $x_t = \sqrt{\alpha_t}x_0 + \sqrt{(1 - \alpha_t)}\epsilon$. The denoising network denoted by $\epsilon_\theta(x_t, c, t)$ is pre-trained to denoise the noisy image $x_t$ to obtain $x_{t-1}$. For better training efficiency, the noising along with the denoising operation occurs in a latent space defined by $z = \mathcal{E}(x)$, where $\mathcal{E}$ is an encoder such as VQ-VAE (van den Oord et al., 2017). Usually, the conditional input $c$ to the denoising network $\epsilon_\theta(.)$ is a text-embedding of the caption $c$ through a text-encoder $c = v_\gamma(c)$. The pre-training objective for diffusion models can be defined as follows:

$$\mathcal{L}(\theta) = \mathbb{E}_{(x_0,c) \sim \mathcal{D}, \epsilon, t} \left[ \|\epsilon - \epsilon_\theta(z_t, c, t)\|_2^2 \right],$$

where $\theta$ is the set of learnable parameters in the UNet $\epsilon_\theta$. During inference, where the objective is to synthesize an image given a text-embedding $c$, a random Gaussian noise $z_T \sim \mathcal{N}(0, I)$ is iteratively denoised for a fixed range of time-steps to produce the final image.

## 2.2 COMPOSITIONALITY EVALUATION METRICS

We focus on the disentangled BLIP-Visual Question Answering (referred to as VQA for simplicity) score proposed by Huang et al. (2023) as a key metric for evaluating image quality. The VQA score measures how accurately an image captures the compositional elements described in the prompt, offering a closer correlation with human judgment compared to metrics like CLIP-Score (Hessel et al., 2021).

## 2.3 DATASET COLLECTION

We utilize the T2I-CompBench dataset (Huang et al., 2023), focusing on three key categories: color, texture, and shape, with a total of 1,000 prompts across both training and evaluation sets. T2I-CompBench is a well-established and widely recognized dataset (Esser et al., 2024). This dataset provides distinct training and evaluation splits for each category, enabling a structured approach to assessing performance. To generate high-quality images, we use three generative models: SD 1.4 (Rombach et al., 2021), DeepFloyd, and SynGen (Rassin et al., 2024), creating 100 samples per prompt with SD 1.4, 60 with DeepFloyd, and 50 with SynGen. This ensures a wide variety of generated images, leveraging each model's strengths. For each prompt, we combined all 210 samples from the three models and selected the top 30 with the highest VQA scores, ensuring the final dataset consisted of images that most accurately reflected the prompts.

## 3 SOURCE (I) : ERRONEOUS ATTENTION CONTRIBUTIONS IN CLIP

In this section, we leverage attention contributions (Elhage et al., 2021; Dar et al., 2023) to analyze the text-embeddings of compositional prompts in the CLIP text-encoder (which is commonly used in many text-to-image models) and compare them with T5-text encoder of DeepFloyd, a model which results in stronger compositionality. Many of the compositional prompts from Huang et al. (2023) have a decomposable template of the form $a_i\ o_j + a_j\ o_j$, where $a_i, a_j$ are attributes (e.g., "black", "matted") while $o_i, o_j$ describe

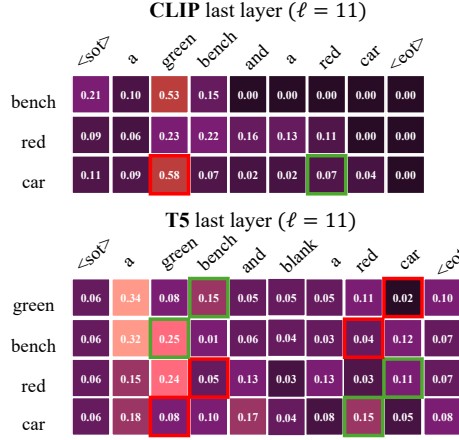

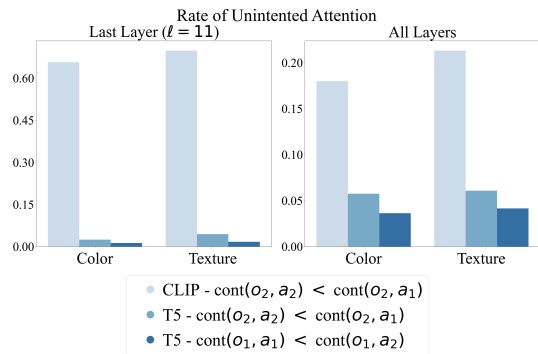

Figure 3: The heatmap illustrates unintended attention contributions in CLIP, while highlighting the more accurate performance of T5.

Figure 4: Quantitatively, we find CLIP to have significantly higher erroneous attention contributions averaged across 780 prompts of color dataset and 582 prompts of texture dataset.

objects (e.g., "car", "bag"). We use attention contributions to understand how the text-embeddings of the compositional tokens (e.g., $\mathbf{a}_i, \mathbf{a}_j, \mathbf{o}_i, \mathbf{o}_j$) are formed for both T5 and CLIP over the layers of these models.

The attention mechanism in layer $\ell$ of a transformer consists of four weight matrices $W_{\mathrm{q}}, W_{\mathrm{v}}, W_{\mathrm{k}}, W_{\mathrm{o}}$ (Vaswani et al., 2017). Each of these weight matrices is divided into $H$ heads denoted by $W_{\mathrm{q}}^h, W_{\mathrm{v}}^h, W_{\mathrm{k}}^h \in \mathbb{R}^{d \times d_h}, W_{\mathrm{o}}^h \in \mathbb{R}^{d_h \times d}$ for all $h \in [H]$. Note that $d_h$ is the dimension of the internal token embeddings. We omit $\ell$ for simplicity, but each layer has its own attention matrices. These matrices are applied on the token embeddings of the output of layer $\ell - 1$, denoted by $\bar{\mathbf{x}}_j$ for token $j$ in that layer. We denote by $\mathrm{q}_j^h, \mathrm{k}_j^h$, and $\mathrm{v}_j^h$ the projection of $\bar{\mathbf{x}}_j$ on query, key, and value matrices of the $h$-th head of layer $\ell$. More precisely,

$$\mathrm{q}_j^h = \bar{\mathbf{x}}_j W_{\mathrm{q}}^h, \quad \mathrm{k}_j^h = \bar{\mathbf{x}}_j W_{\mathrm{k}}^h, \quad \mathrm{v}_j^h = \bar{\mathbf{x}}_j W_{\mathrm{v}}^h.$$

The *contribution* of token $j$ to token $i$ in layer $\ell$, denoted by $\mathrm{cont}_{i,j}$, is computed as follows:

$$\mathrm{cont}_{i,j} = \left\| \sum_{h=1}^{H} \mathrm{attn}_{i,j}^h \, \mathrm{v}_j^h \, W_{\mathrm{o}}^h \right\|_2$$

where $\mathrm{attn}_{i,j}^h$ is the attention weight of token $i$ to $j$ in the $h$-th head of layer $\ell$. Specifically,

$$\mathrm{attn}_{i,\cdot}^h = \mathrm{SOFTMAX} \left( \left\{ \frac{\langle \mathrm{q}_i^h, \mathrm{k}_j^h \rangle}{\sqrt{d_h}} \right\}_{j=1}^{n} \right).$$

Notably, $\mathrm{cont}_{i,j}$ is a significant metric that quantifies the *contribution* of a token $j$ to the norm of a token $i$ at layer $\ell$. We employ this metric to identify layers in which important tokens highly attend to *unintended* tokens, or lowly attend to *intended* ones. We refer to Appendix C.1 for more details on attention contribution.

### 3.1 KEY FINDING: T5 HAS LESS ERRONEOUS ATTENTION CONTRIBUTIONS THAN CLIP

We refer to Figure 3 that visualizes attention contribution of both T5 and CLIP text-encoder in the last layer ($\ell = 11$) for the prompt "a green bench and a red car". Ideally, the attention mechanism should guide the token

"car" to focus more on "red" than "green", but in the last layer of the CLIP text-encoder, "car" significantly attends to "green". In contrast, T5 shows a more consistent attention pattern, with "red" contributing more to the token "car" and "green" contributing more to the token "bench".

We further conduct an extensive analysis on specific types of prompts, consisting of 780 prompts of color dataset and 582 prompts of texture dataset, each structured as "$\mathbf{a}_1$ $\mathbf{o}_1$ and $\mathbf{a}_2$ $\mathbf{o}_2$." For each prompt, we obtain attention contributions in all layers and count the number of layers where *unintended attention contributions* occur. In the CLIP text-encoder, unintended attention occurs when $\mathbf{o}_2$ attends more to $\mathbf{a}_1$ than $\mathbf{a}_2$. For T5, it occurs when $\mathbf{o}_2$ attends more to $\mathbf{a}_1$ than $\mathbf{a}_2$, or $\mathbf{o}_1$ attends more to $\mathbf{a}_2$ than $\mathbf{a}_1$. Figure 4 provides a quantitative comparison of unintended attention across various prompts between the CLIP text-encoder and T5. The T5 model demonstrates improved performance on our metric compared to the CLIP text-encoder, reinforcing the hypothesis that erroneous attention mechanisms in CLIP may contribute to its weaker compositionality in text-to-image models. This aligns with the general observation that pretrained text-to-image models using the T5 text-encoder tend to exhibit superior compositionality. Additional details can be found in Appendix C.4. Further experiments with other text-encoders are also reported in Appendix C.3.

### 3.2 ZERO-SHOT ATTENTION REWEIGHTING

Inspired by attention mechanism shortcomings of CLIP text-encoder, we aim to improve compositionality of CLIP-based diffusion models by zero-shot reweighting of the attention maps. Specifically, we apply a hand-crafted zero-shot manipulation of the attention maps in certain layers of the CLIP text-encoder to effectively reduce unintended attentions while enhancing meaningful ones. This zero-shot reweighting is applied to the logits before the SOFTMAX layer in the last three layers of the text-encoder. More precisely, we compute a matrix $M \in \mathbb{R}^{n \times n}$ and add it to the attention logits. For each head $h$, the new attention values are computed and then propagated through the subsequent layers of the text encoder:

$$\text{attn}'^{h}_{i,:} = \text{SOFTMAX}\left( \left\{ \frac{\langle \mathbf{q}^h_i, \mathbf{k}^h_j \rangle}{\sqrt{d_h}} + M_{i,j} \right\}^n_{j=1} \right).$$

We set the values in $M$ by considering the ideal case where no incorrect attentions occur in the mechanism. For example, for prompt "a green bench and a red car", we ensure that the token "car" does not attend to the token "green" by assigning a sufficiently large negative value to the corresponding entry in matrix $M$. Further details on how we obtain matrix $M$ can be found in Appendix C.2.

**Key Results.** Applying zero-shot attention reweighting with matrix $M$ on 780 compositional prompts of color dataset, we achieved a 2.93% improvement in VQA scores. Examples of effective zero-shot reweighting, demonstrating its impact on mitigating compositionality issues in can be found in Appendix C.2. Although erroneous attention contributions in the CLIP text-encoder is one source of error, it is not the primary error source due to modest improvements in compositional accuracy. In the next section, we investigate the sub-optimality of the output space of CLIP text-encoder, which we find to be a significant source of error.

## 4 SOURCE (II) : SUB-OPTIMALITY OF CLIP TEXT-ENCODER FOR COMPOSITIONAL PROMPTS

In this section, we understand if the UNet is capable of generating compositional scenes by optimizing the text-embeddings that it takes as the conditional input. Given an input prompt $c$ with a particular composition (e.g., *"a red book and a yellow table"*), we utilize our dataset and obtain $\mathcal{D}_c$ including high-quality compositional images for prompt $c$. We then optimize the output text-embedding $\mathbf{c}$ as follows:

$$\mathbf{c}^* = \arg\min_{\mathbf{c}} \mathbb{E}_{x_0 \sim \mathcal{D}_c, \epsilon, t} \left[ \|\epsilon - \epsilon_\theta(\mathbf{z}_t, \mathbf{c}, t)\|^2_2 \right].$$

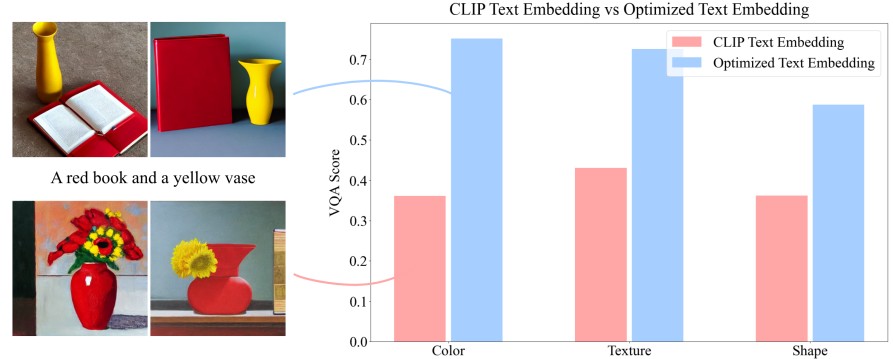

Figure 5: Comparative analysis of VQA Scores between CLIP text-embeddings and optimized text-embeddings using Stable Diffusion v1.4 across color, texture, and shape categories. Results show CLIP text embeddings achieved scores of 0.3615 for color, 0.4306 for texture, and 0.3619 for shape, while optimized text embeddings achieved scores of 0.7513 for color, 0.7254 for texture, and 0.58728 for shape.

We then use $\mathbf{c}^*$ to generate images using the UNet $\epsilon_\theta$ across different seeds. Figure 5 depicts a few of generated images using optimized text-embeddings.

**Key Results.** As seen in Figure 5, we consistently improve VQA scores across a variety of compositional prompts (i.e., color, texture, and shape). This indicates that CLIP text-encoder does not output the proper text-embedding suitable for generating compositional scenes. However, that optimized embedding space exists, highlighting the ability of UNet to generate coherent compositional scenes when a proper text-embedding is given. This further motivates the idea of improving CLIP output space to mitigate compositionality issues in text-to-image diffusion models. We refer to Appendix B for other configurations showing that optimizing a subset of tokens can also effectively improve compositionality.

## 5 LINEAR PROJECTION ON CLIP: A SIMPLE BASELINE TO IMPROVE COMPOSITIONALITY IN TEXT-TO-IMAGE GENERATIVE MODELS

In this Section, we provide two baselines `CLP` and `WiCLP` that are linear modification of CLIP output to map that sub-optimal space to an enhanced one, better suited for compositionality.

### 5.1 `CLP`: TOKEN-WISE COMPOSITIONAL LINEAR PROJECTION

Given the text-embedding $\mathbf{c} \in \mathbb{R}^{n \times d}$ as the output of the text-encoder for prompt $c$, i.e., $\mathbf{c} = v_\gamma(c)$, we train a linear projection $\mathtt{CLP}_{W,b} : \mathbb{R}^{n \times d} \to \mathbb{R}^{n \times d}$. This projection includes a matrix $W \in \mathbb{R}^{d \times d}$ and a bias term $b \in \mathbb{R}^d$, which are applied token-wise to the output text-embeddings of the encoder. More formally, for $\mathbf{c} \in \mathbb{R}^{n \times d}$ including text-embeddings of $n$ tokens $\mathbf{c}_1, \mathbf{c}_2, \cdots, \mathbf{c}_n \in \mathbb{R}^d$, $\mathtt{CLP}_{W,b}(\mathbf{c})$ is obtained by stacking projected embeddings $\mathbf{c}'_1, \mathbf{c}'_2, \cdots, \mathbf{c}'_n$ where $\mathbf{c}'_i = W^T \mathbf{c}_i + b$.

Finally, we solve the following optimization problem on a dataset $\mathcal{D}$ including image-caption pairs of high-quality compositional images:

$$W^*, b^* = \arg\min_{W,b} \mathbb{E}_{(x_0,c) \sim \mathcal{D}, \epsilon, t} \left[ \left\| \epsilon - \epsilon_\theta \left( \mathbf{z}_t, \mathtt{CLP}_{W,b}(\mathbf{c}), t \right) \right\|_2^2 \right].$$

We then apply $\mathtt{CLP}_{W^*,b^*}$ on CLIP text-encoder to obtain improved embeddings.

prompt: "A bathroom with green tile and a red shower curtain"

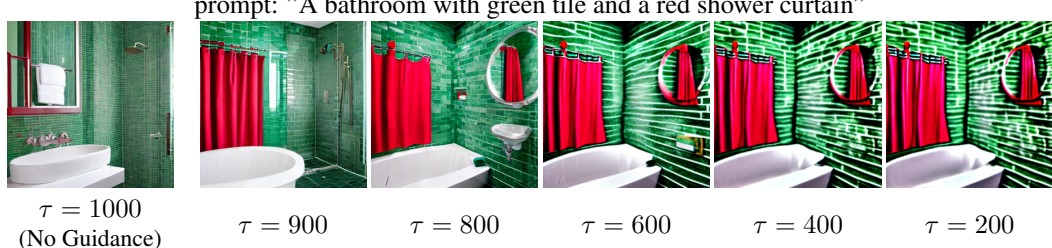

| $\tau = 1000$ (No Guidance) | $\tau = 900$ | $\tau = 800$ | $\tau = 600$ | $\tau = 400$ | $\tau = 200$ |

Figure 6: Qualitative results showing the impact of SWITCH-OFF with varying thresholds $\tau$.

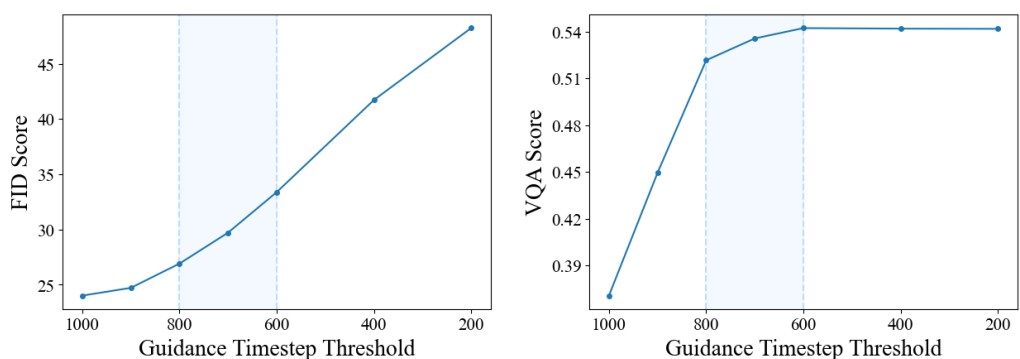

Figure 7: Trade-off between VQA and FID scores with SWITCH-OFF at different thresholds.

## 5.2 WiCLP: WINDOW-BASED COMPOSITIONAL LINEAR PROJECTION

In this section, we propose a more advanced linear projection scheme where the new embedding of a token is derived by applying a linear projection on that token in conjunction with a set of its adjacent tokens, i.e., tokens within a specified window. This method not only leverages the benefits of CLP but also incorporates the contextual information from neighboring tokens, potentially leading to more precise text-embeddings.

More formally, we train a mapping $\text{WiCLP}_{W,b} : \mathbb{R}^{n \times d} \to \mathbb{R}^{n \times d}$ including a parameter $s$ (indicating window length), matrix $W \in \mathbb{R}^{(2s+1)d \times d}$, and a bias term $b \in \mathbb{R}^d$. For text-embeddings $\mathbf{c} \in \mathbb{R}^{n \times d}$ consisting of $n$ token embeddings of $\mathbf{c}_1, \mathbf{c}_2, \cdots, \mathbf{c}_n \in \mathbb{R}^d$, we obtain $\text{WiCLP}_{W,b}$ by stacking projected embeddings $\mathbf{c}'_1, \mathbf{c}'_2, \cdots, \mathbf{c}'_n$ where

$$\mathbf{c}'_i = W^T \text{ CONCATENATION} \left( (\mathbf{c}_j)_{j=i-s}^{i+s} \right) + b$$

Similarly, we solve the following optimization problem to train the projection:

$$W^*, b^* = \arg \min_{W,b} \mathbb{E}_{(x_0,c) \sim \mathcal{D}, \epsilon, t} \left[ \left\| \epsilon - \epsilon_\theta \left( \mathbf{z}_t, \text{WiCLP}_{W,b} \left( \mathbf{c} \right), t \right) \right\|_2^2 \right].$$

Note that we use $s = 2$, i.e., window length of $5$ in our experiments.

**Comparison between CLP and WiCLP.** We observe that WiCLP improves over CLP (special case of WiCLP with $s = 0$) by incorporating adjacent tokens in addition to the actual token. This approach likely improves embeddings by mitigating unintended attention from adjacent tokens. For discussion on choosing the window length ($s$) in WiCLP, see Appendix D.6.

| Model | | Color | Texture | Shape |
|---|---|---|---|---|
| Stable Diffusion v1.4 | Baseline | 0.3765 | 0.4156 | 0.3576 |
| | `CLP` | 0.4837 | 0.5312 | 0.4307 |
| | `WiCLP` | **0.5383** | **0.5671** | **0.4527** |
| Stable Diffusion v2 | Baseline | 0.5065 | 0.4922 | 0.4221 |
| | Composable (Liu et al., 2022b) | 0.4063 | 0.3645 | 0.3299 |
| | Structured (Feng et al., 2022) | 0.4990 | 0.4900 | 0.4218 |
| | Attn-Exct (Chefer et al., 2023a) | 0.6400 | 0.5963 | 0.4517 |
| | GORS-unbaised (Huang et al., 2023) | 0.6414 | 0.6025 | 0.4546 |
| | `CLP` | 0.6075 | 0.5707 | 0.4567 |
| | `WiCLP` | **0.6500** | **0.6036** | **0.4821** |

Table 1: Quantitative comparison with state-of-the-art and baseline methods across different categories of the T2I-CompBench dataset

### 5.3 SWITCH-OFF: TRADE-OFF BETWEEN COMPOSITIONALITY AND CLEAN ACCURACY

Fine-tuning models or adding modules to a base model often results in a degradation of image quality and an increase in the Fréchet Inception Distance (FID) score. To balance the trade-off between improved compositionality and the quality of generated images for clean prompts – an important issue in existing work – inspired by Hertz et al. (2022), we adopt SWITCH-OFF, where we apply the linear projection only during the initial steps of inference. Specifically, given a time-step threshold $\tau$, for $t \geq \tau$, we use $\text{WiCLP}_{W^*,b^*}(\mathbf{c})$, while for $t < \tau$, we use the unchanged embedding $\mathbf{c}$ as the input to the cross-attention layers.

Figure 7 illustrates the trade-off between VQA score and FID on a randomly sampled subset of MS-COCO (Lin et al., 2014) for different choices of $\tau$. As shown, even a large value of $\tau$ suffices for obtaining high-quality compositional scenes as the composition of final generated image is primarily formed at early steps. Thus, choosing a large $\tau$ preserves the model's improved compositionality while maintaining its clean accuracy. Setting $\tau = 800$ offers a competitive VQA score compared to the model where projection is applied at all time steps, and achieves a competitive FID similar to that of the clean model. Figure 6 depicts a few images generated using different choices of $\tau$. We refer to Appendix D.5 for more visualizations.

## 6 EXPERIMENTS

**Existing Baselines.** We evaluate the performance of four methods alongside standard models SD v1.4 and SD v2. These include Composable Diffusion (Liu et al., 2022b), which addresses concept conjunction and negation in pretrained diffusion models; Structured Diffusion (Feng et al., 2022), which focuses on attribute binding; Attn-Exct (Chefer et al., 2023a), which ensures correct attention to all subjects in the prompt; and GORS (Huang et al., 2023), which fine-tunes Stable Diffusion v2 using a reward function. GORS optimizes more parameters but underperforms slightly compared to our method, while Attn-Exct requires iterative optimizations during inference, making it slower than our method, which adds only a linear projection layer.

**Training Setup.** All of the models are trained using the objective function of diffusion models on color, texture, and shape datasets. During training, we keep all major components frozen, including the U-Net, CLIP text-encoder, and VAE encoder and decoder, and only the linear projections are trained. We refer to Appendix D.1 for details on the training procedure.

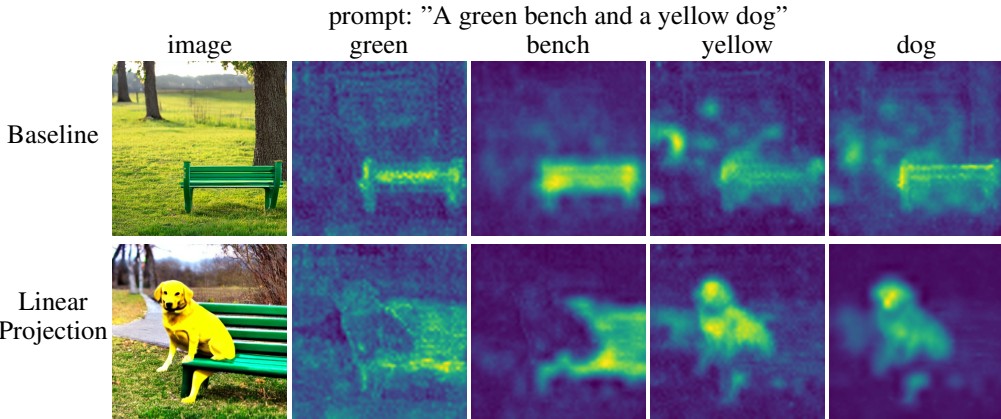

Figure 8: Applying `CLP` results in more accurate cross-attention maps.

## 6.1 QUALITATIVE AND QUANTITATIVE EVALUATION

**Qualitative Evaluation.** Figure 2 presents images generated when applying `CLP` and `WiCLP`. When generating compositional prompts with a baseline model, objects are often missing or attributes are incorrectly applied. However, with `CLP` and `WiCLP`, objects and their corresponding attributes are more accurately generated. We refer to Appendix D.3 for more visualizations. Figure 8 illustrates cross-attention maps for a sample prompt. In the base model, attention maps are flawed, with some tokens incorrectly attending to the wrong pixels. However, with both `CLP` and `WiCLP`, objects and attributes more accurately attend to their respective pixels. For more visualizations, see Appendix D.4.

**Quantitative Evaluation.** VQA scores of our method and other discussed baselines are provided in Table 1. As shown, both `CLP` and `WiCLP` significantly improve upon the baselines. `WiCLP` achieves higher VQA scores compared to other state-of-the-art methods, despite its simplicity. Interestingly, our improved results do not compromise the model's general utility. Our method causes a slight increase in FID score on MS-COCO prompts compared to base models, but this increase is smaller than other baselines—for example, `WiCLP` scores 27.40 versus GORS at 30.54. Further FID performance details are available in Appendix D.2.

**Human Experiments.** We conducted a human evaluation where participants compared images generated by SD v1.4 and SD v1.4 + `WiCLP`, selecting the image that best matched the given prompt. The results showed that in $34.625\%$ of cases, evaluators chose the base model's image; in $51.875\%$, they preferred the `WiCLP` images; and in $13.50\%$, they rated both equally. Further details can be found in Appendix D.2.

## 7 CONCLUSION

Our paper examines potential error sources in text-to-image models for generating images from compositional prompts. We identify two error sources: (i) A minor error source, where the token embeddings in the CLIP text-encoder have erroneous attention contributions and (ii) A major error source, where we find the output space of the CLIP text-encoder to be sub-optimally aligned to the UNet for compositional prompts. Leveraging our observations, we propose a simple and strong baseline `WiCLP` which involves fine-tuning a linear projection on CLIP's representation space. `WiCLP` though inherently simple and parameter efficient, outperforms existing methods on compositional image generation benchmarks and maintains a low FID score on a broader range of clean prompts. We discuss limitations in Appendix A.

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

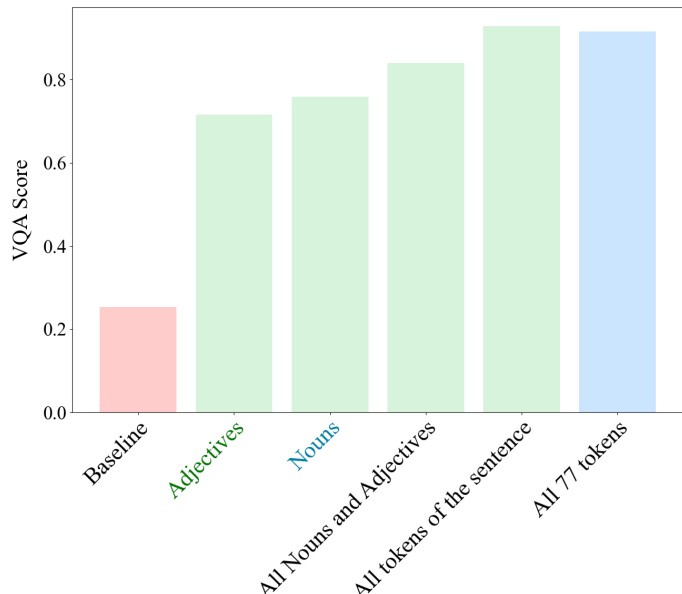

Figure 9: Comparison of VQA scores when optimizing different subsets of tokens for the sample prompt: "A red book and a yellow vase"

## A    LIMITATIONS

In this paper, we have thoroughly analyzed one of the key reasons why Stable Diffusion struggles to generate compositional prompts and proposed a lightweight method to mitigate this issue. However, there remains significant room for improvement in this area. Our approach focuses on improving the text encoder, which we identified as a major source of error. There are potentially other sources of the issue within the entire generative model pipeline that need to be explored. Additionally, our method involves a small fine-tuning step using a simple linear projection. Future work could explore alternative approaches, such as more sophisticated fine-tuning techniques, advanced attention mechanisms, or hybrid models that integrate multiple strategies.

## B    OPTIMIZING THE TEXT-EMBEDDINGS OF A SUBSET OF TOKENS

Given $\mathbf{c} \in \mathbb{R}^{n \times d}$, where $n$ refers to the number of tokens and $d$ refers to the dimensionality of the text-embedding, for the second configuration we only optimize a subset of tokens $n' \in n$. We refer to this subset of tokens as $\mathbf{c}'$. These tokens correspond to relevant parts of the prompt which govern compositionality (e.g., "red book" and "yellow table" in "A red book and an yellow table").

$$\mathbf{c}'^* = \arg\min_{\mathbf{c}'} \mathbb{E}_{\epsilon,t} ||\epsilon - \epsilon_\theta(\mathbf{z}_t, \mathbf{c}', t)||_2^2,$$

Figure 9 shows the results for the sample prompt "a red book and a yellow vase". We considered different subsets of tokens $n'$: adjectives ("red" and "yellow"), nouns ("book" and "vase"), both nouns and adjectives, and all tokens in the sentence. The results indicate that optimizing even a few tokens significantly improves the VQA score. However, optimizing all tokens in the sentence yields the highest score.

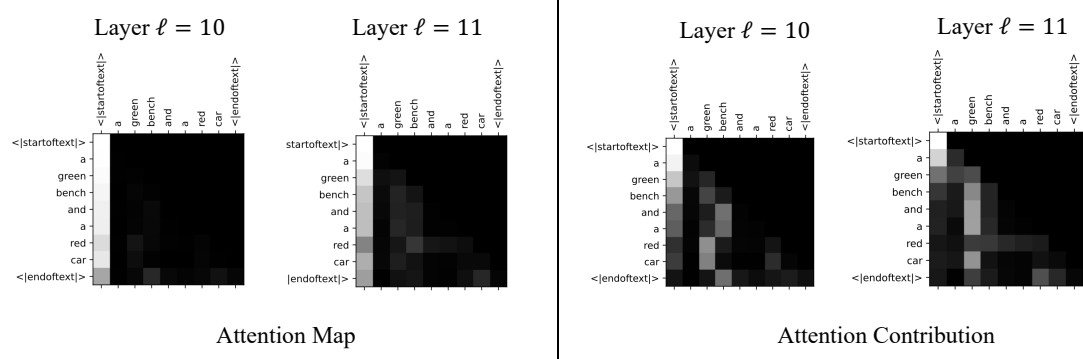

Figure 10: Visualization of attention map and attention contribution for prompt "a green bench and a red car" over different layers of CLIP. Contribution provides better insight on the attention mechanism.

## C    Source (i) : Erroneous Attention Contributions

### C.1    Attention Contribution

In this Section, we provide more details on our analysis to quantitatively measure tokens' contribution to each other in a layer of attention mechanism. One natural way of doing this analysis is to utilize attention maps $\text{attn}_{i,j}^h$ and aggregate them over heads, however, we observe that this map couldn't effectively show the contribution. Attention map does not consider norm of tokens in the previous layer, thus, does not provide informative knowledge on how each token is formed in the attention mechanism. In fact, as seen in Figure 10, we cannot obtain much information by looking at these maps while attention contribution clearly shows amount of norm that comes from each of the attended tokens.

### C.2    Zero-shot Attention Reweighting

To fix unintended attentions, we aim to compute a matrix $M$ to be applied across various heads in the last few layers of CLIP, reducing the effect of wrong attention, leading to more accurate text-embeddings that are capable of generating high-quality compositional scenes. To avoid unintended attention for prompts of the form "$\mathbf{a}_1\mathbf{o}_1 + \mathbf{a}_2\mathbf{o}_2$", we add large negative values to entries $M_{\mathbf{o}_2,\mathbf{a}_1}$, $M_{\mathbf{a}_2,\mathbf{a}_1}$, and some positive value to $M_{\mathbf{o}_2,\mathbf{a}_2}$ and $M_{\mathbf{o}_1,\mathbf{a}_1}$, and small negative value to $M_{\mathbf{o}_2,\mathbf{o}_1}$. To find what values to assign to those entries, we consider a small set of prompts in color dataset (5 prompts in total) and obtain parameters for that matrix to maximize VQA score. Figure 11 shows few examples of zero-shot modification.

### C.3    Experiments with LLaMa3 8B

We explored the analysis of attention contributions to identify unintended attention in LLaMa3 8B, which utilizes a more advanced text encoder specifically designed for language modeling and pretrained on large-scale text corpora. Table 2 reports the rate of unintended attention across prompts in the color and texture datasets. The results demonstrate that unintended attention occurs less frequently in more advanced text encoders, further emphasizing the limitations of the CLIP text encoder.

|  | color | | texture | |
| --- | --- | --- | --- | --- |
|  | last layer | all layers | last layer | all layers |
| LLaMa3 | 0.015 | 0.081 | 0.033 | 0.066 |
| CLIP | 0.657 | 0.187 | 0.696 | 0.213 |

Table 2: Unintended attention rate in LLaMa3 8B vs CLIP. LLaMa3 shows significant less unintended attentions.

|  | SD v1.4 | SD v2 | SD v1.4 + `WiCLP` | SD v2 + `WiCLP` | GORS |
| --- | --- | --- | --- | --- | --- |
| FID Score | 24.33 | 23.27 | 25.40 | 27.40 | 30.54 |

Table 3: Comparison of FID scores between the baseline models and `WiCLP` using SWITCH-OFF with $\tau = 800$, as well as the GORS approach.

## C.4 MODELS WITH T5 TEXT-ENCODER

We conducted experiments to measure the VQA score on the color dataset for models that use T5 as their text encoder. DeepFloyd achieved a score of $0.604$, which is significantly higher than that of SD-v1.4. Additionally, DeepFloyd-I-M, which employs a smaller first-stage UNet compared to DeepFloyd, obtained a score of $0.436$, also surpassing the SD-v1.4 score.

## D EXPERIMENTS

### D.1 TRAINING SETUP

In this section, we present the details of the experiments conducted to evaluate our proposed methods. The training is performed for 25,000 steps with a batch size of 4. An RTX A5000 GPU is used for training models based on Stable Diffusion 1.4, while an RTX A6000 GPU is used for models based on Stable Diffusion 2. We employed the Adam optimizer with a learning rate of $1 \times 10^{-5}$ and utilized a Multi-Step learning rate scheduler with decays ($\alpha = 0.1$) at 10,000 and 16,000 steps. For the `WiCLP`, a window size of 5 was used. All network parameters were initialized to zero, leveraging the skip connection to ensure that the initial output matched the CLIP text embeddings. Our implementation is based on the Diffusers[2] library, utilizing their modules, models, and checkpoints to build and train our models. This comprehensive setup ensured that our method was rigorously tested under controlled conditions, providing a robust evaluation of its performance.

### D.2 EXTENDED EVALUATION

**Human Evaluation** We conducted a human evaluation in which participants compared images generated by SD v1.4 and SD v1.4 + `WiCLP`, selecting the image that best matched the given prompt (Figure 18). Five evaluators were presented with 200 randomly selected image pairs, evaluating a total of 1000 image-caption pairs.

**TIFA Metric.** To provide a more comprehensive evaluation, in addition to the disentangled BLIP-VQA score proposed by Huang et al. (2023), we also incorporate the TIFA metric (Hu et al., 2023). TIFA (Text-to-Image Faithfulness Evaluation with Question Answering) is an automated evaluation method that measures how faithfully a generated image corresponds to its textual input via visual question answering (VQA). It generates

---

[2]https://github.com/huggingface/diffusers

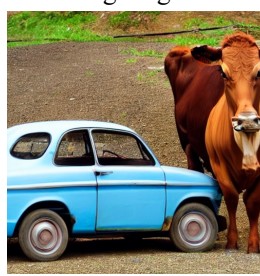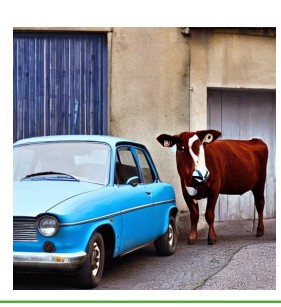
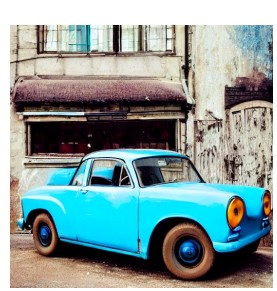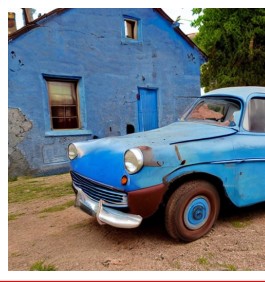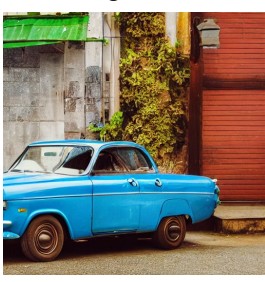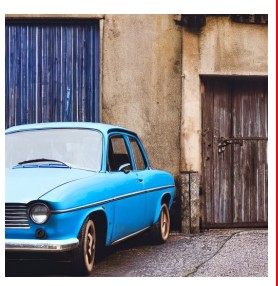

Figure 11: Visualization of some images generated with same set of seeds using original text-embeddings of prompt "a blue car and a brown cow" and text-embeddings that are obtained as the result of zero-shot reweighting of attention matrix.

multiple question-answer pairs from the text input using a language model, then evaluates image faithfulness by determining whether existing VQA models can accurately answer these questions based on the image. As a reference-free metric, TIFA offers fine-grained and interpretable assessments of image quality.

Using TIFA, we observed that SD v1.4 and SD v2 achieved scores of 0.6598 and 0.7735, respectively. Notably, the scores for `WiCLP` applied on top of SD v1.4 and SD v2 improved to 0.7462 and 0.8133, respectively, demonstrating the enhanced performance of our approach.

**FID Score Comparison** Our method results in a modest increase in FID score on MS-COCO prompts compared to the base models, as shown in Table 3. However, this increase is less pronounced than in other baselines—for example, SD v2 + `WiCLP` scores 27.40, whereas GORS reaches 30.54.

### D.3  CLP AND `WiCLP` VISUALIZATION

In this section, we provide additional visualizations comparing `CLP`, `WiCLP`, and baseline models in Figures 14, 15.

### D.4  VISUALIZATION OF CROSS-ATTENTIONS

In this section, we provide additional cross-attention map visualizations in Figures 14 and 15.

## D.5 VISUALIZATION OF SWITCH-OFF

In this section, we present more qualitative samples illustrating the effect of SWITCH-OFF at different timestep thresholds for various prompts in Figures 16 and 17.

## D.6 CHOICE OF WINDOW LENGTH IN WiCLP

One might suggest that instead of using token-wise linear projection (CLP) or a window-based linear projection with a limited window (WiCLP), employing a linear projection that considers all tokens when finding a better embedding for each token might yield better results. However, our thorough quantitative study and experiments tested various window sizes for WiCLP. We found that using a window size of 5 achieves the highest performance.

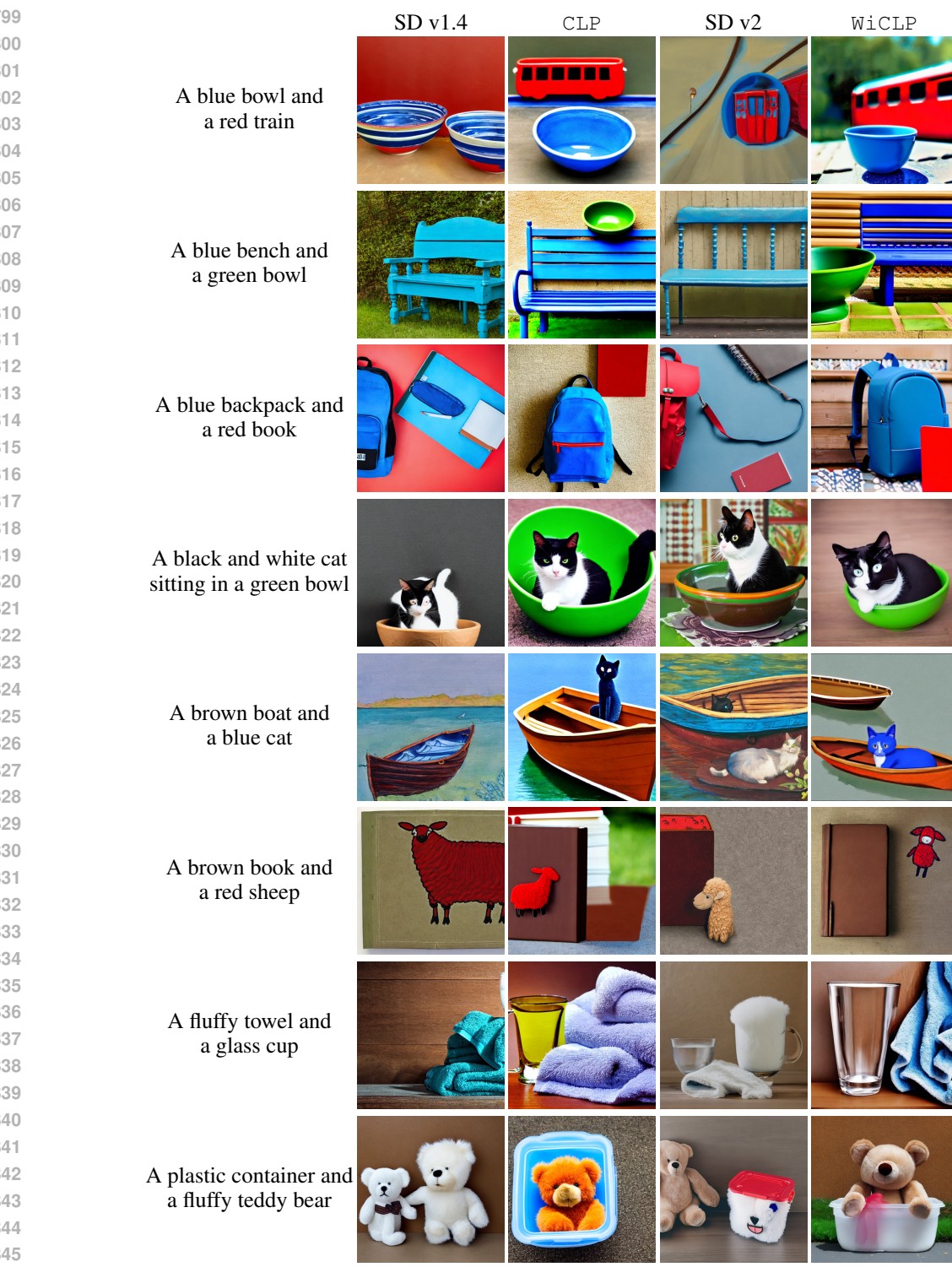

Figure 12: Caption

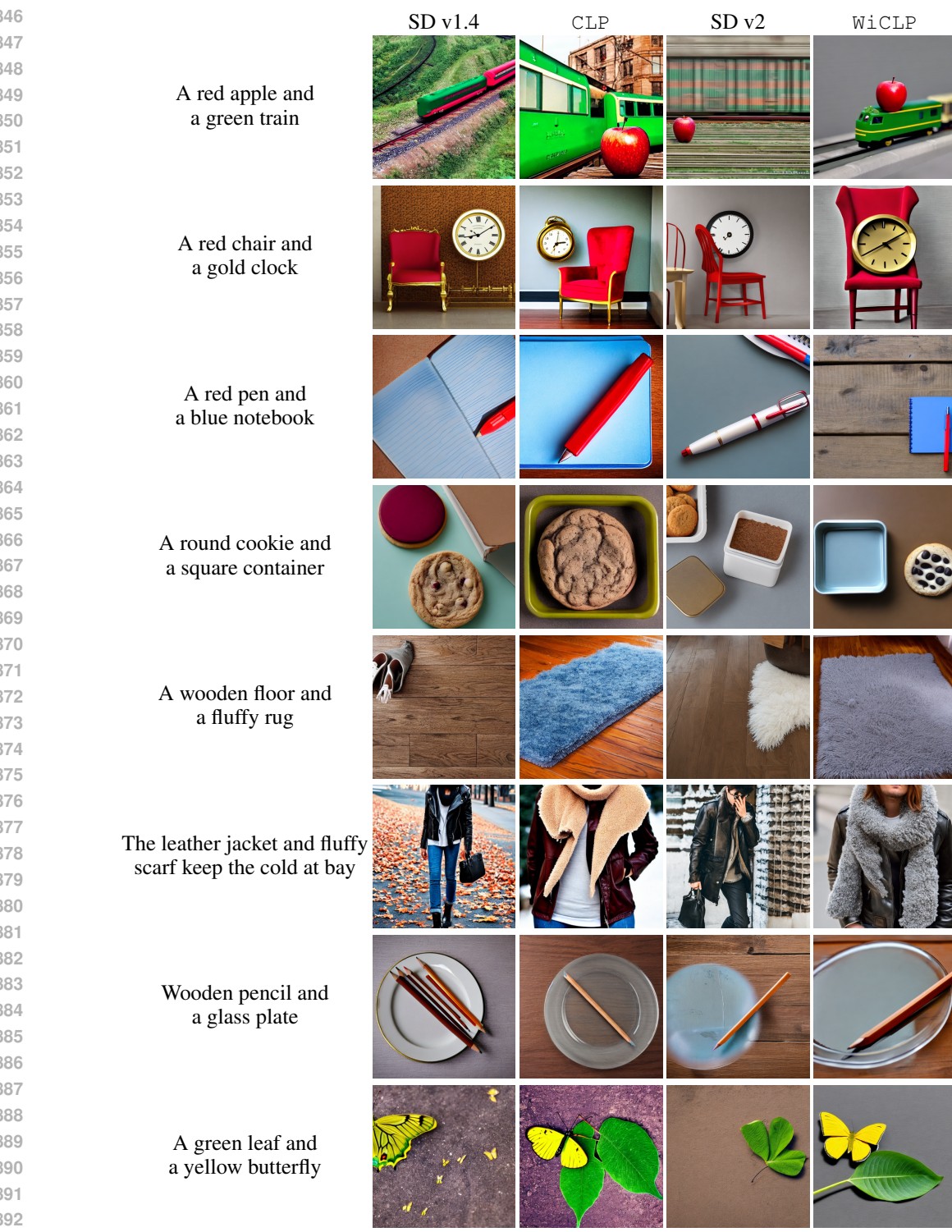

Figure 13: Caption

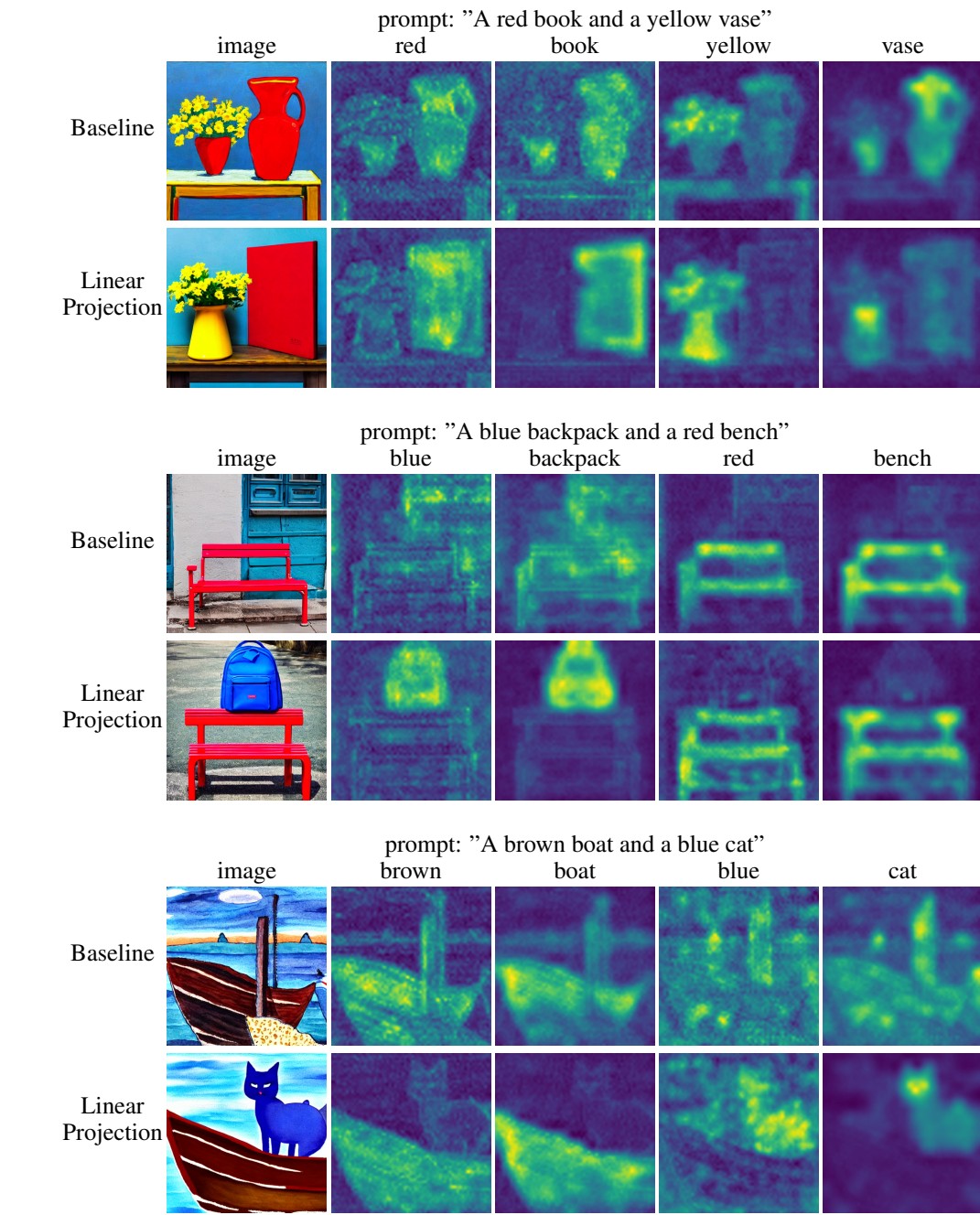

Figure 14: Comparison of cross-attention maps of the U-Net with and without the `CLP`

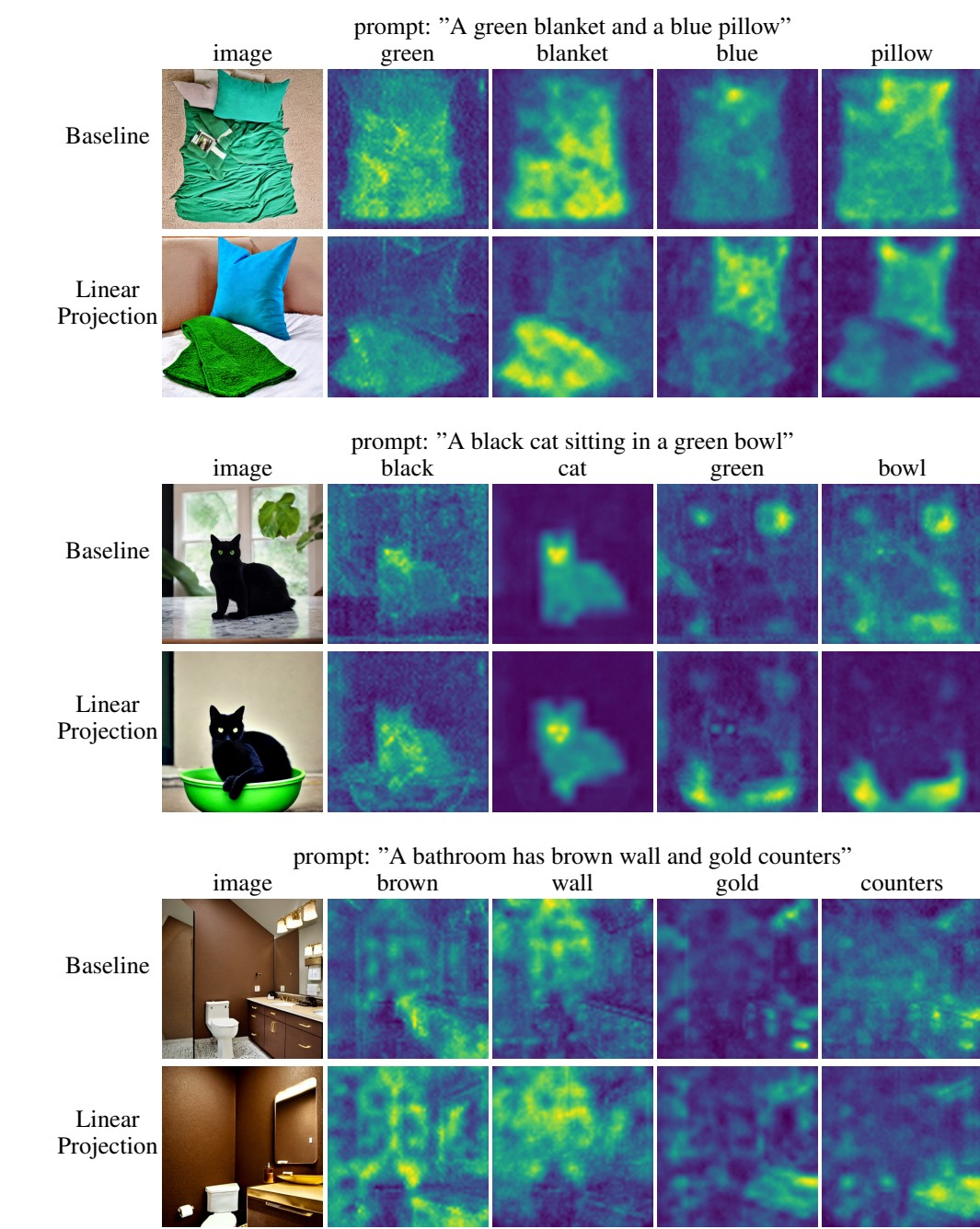

Figure 15: Comparison of cross-attention maps of the U-Net with and without the `CLP`

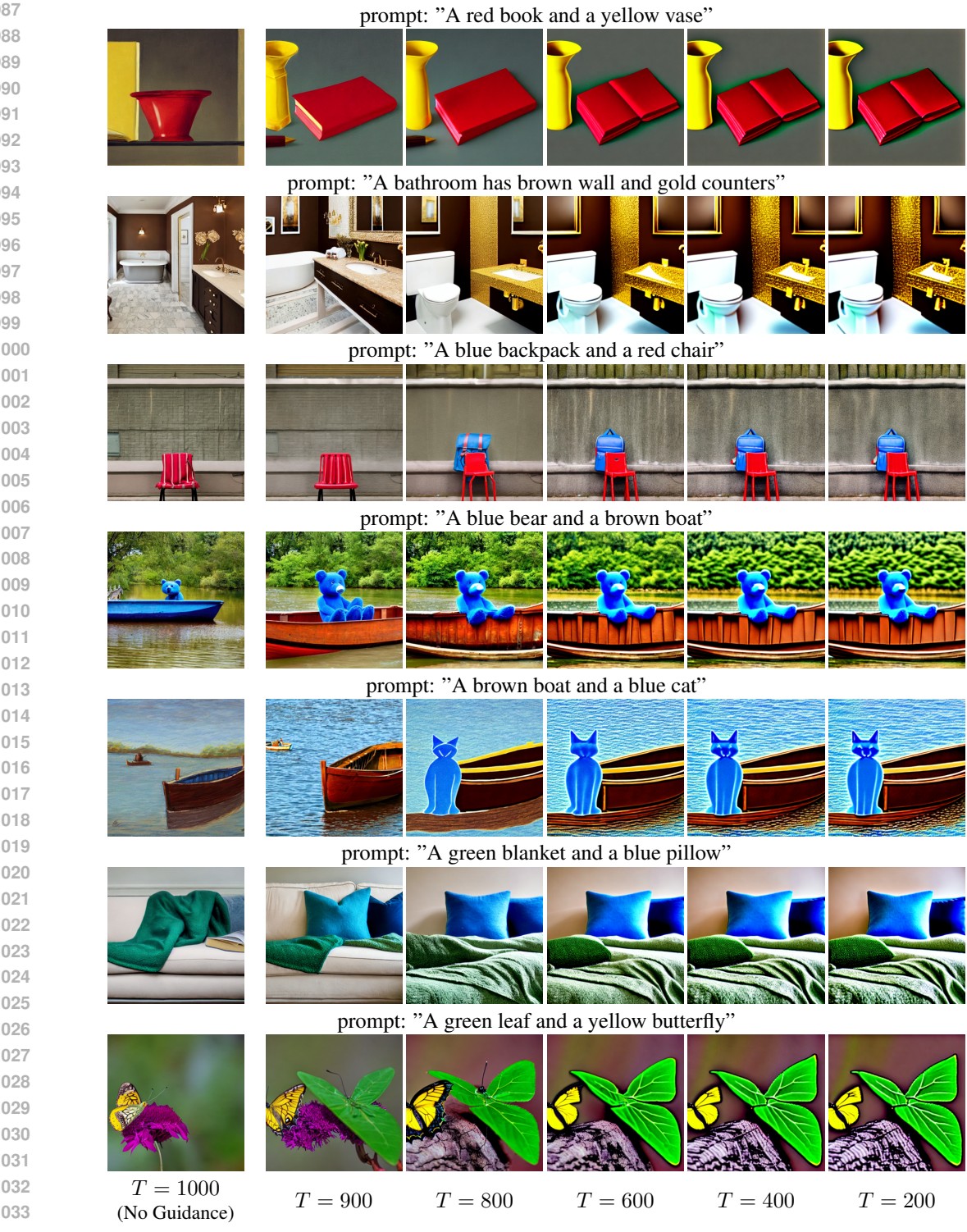

Figure 16: Qualitative results showing the impact of SWITCH-OFF with varying thresholds $T$

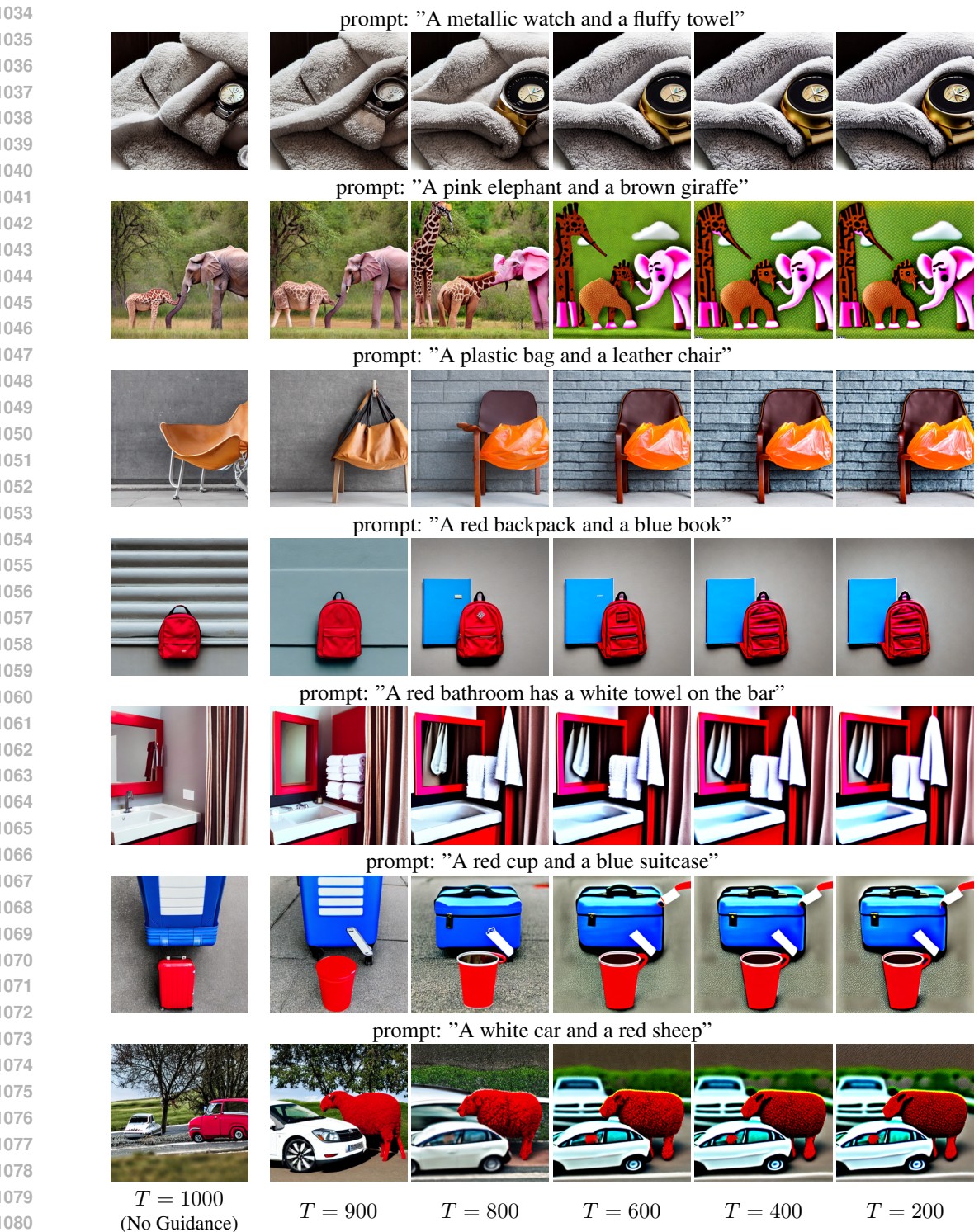

Figure 17: Qualitative results showing the impact of SWITCH-OFF with varying thresholds $T$

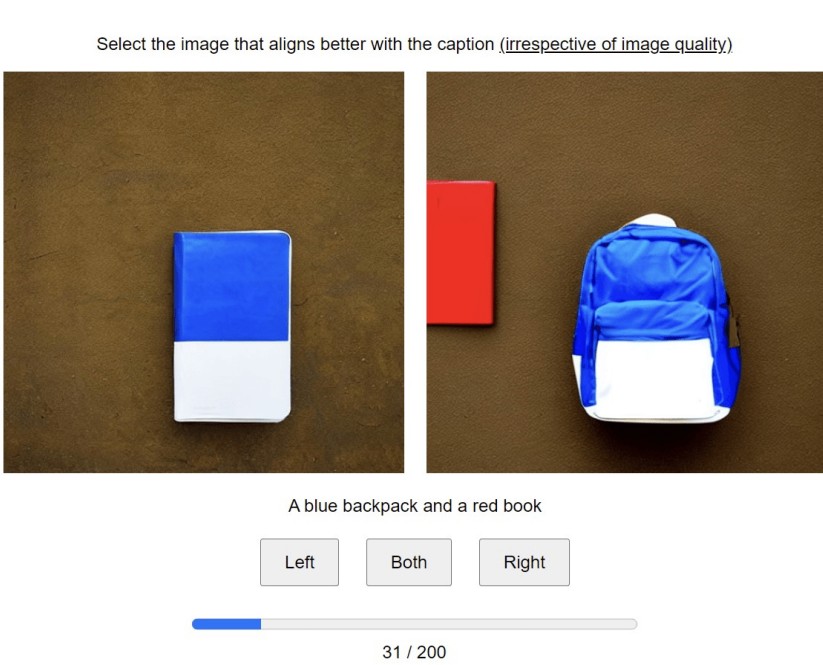

Figure 18: A sample from the human evaluation study, where participants were presented with a pair of images and a caption. They were asked to select the image that best represented the caption or choose 'both' if the images equally captured the caption's meaning.

