# OpenReview forum: "Mitigating Compositional Issues in Text-to-Image Generative Models via Enhanced Text Embeddings"
_ICLR.cc/2025/Conference — ICLR 2025 Conference Withdrawn Submission_

### Official Review · Reviewer_tRxw · 2024-10-30

**Soundness:** 2
**Presentation:** 3
**Contribution:** 2
**Rating:** 5
**Confidence:** 4

**Summary:**

This paper proposes a fine-tuning approach to address the compositionality issue of text-to-image diffusion models. Specifically, it first identifies one of the major sources of the problem is that the attention module in text encoder tend to assign wrong attention of unintended tokens to other tokens, resulting in distorted representation. Then it proposes a window-based projection layer - by learning to aggregate information of tokens within a local window for a new representation. Experimental results show that the proposed method have better compositional alignment with respect to the text description without harming the model's FID score.

**Strengths:**

1. The paper is well motivated, as correctly composing the object / attribute relations in text-to-diffusion models is a very important problem.
2. The paper makes the observation that one source of wrong composition is in the attention assignment operation within the text encoder, which makes intuitive sense and is supported by experimental results.
3. The paper proposes to add a window-based linear projection layer on top of the text embedding for fine tuning, showing better compositionality alignment, while maintaining FID values.

**Weaknesses:**

1. The window-based projection design is based on the assumption that tokens with close semantic relations occur within a local window. However, this is not necessarily the case in more complex sentence, in which semantically related words have longer distance than the length of window, e.g. "three cubes, the color of which are red, green, blue".  So the proposed solution might not fundamentally address the attention assignment issue, which requires a more general sentence parsing method.
2.  There are no failure case visualizations showing under what situations will the proposed method fail to capture the correct relations in the text description.

**Questions:**

1. As mentioned in weaknesses, it would be better to show more generated samples under complex descriptions, to see whether the proposed design is generally applicable. (Specifically, complex in the sense that semantically related words have distance longer than the length of the window)
2. Whether there are examples in the tested datasets that the proposed method failed to generate the expected compositions?

---

> ### Author Response · Authors · 2024-11-22
>
> We thank Reviewer tRxw for their time and thoughtful feedback. Below, we address each of their comments individually:
>
> > The window-based projection design is based on the assumption that tokens with close semantic relations …
>
> The reviewer raises an excellent point. We agree that the window length determines the extent to which a token can access adjacent tokens before the linear projection. However, it is important to note that the CLIP text encoder inherently processes information from all preceding tokens, meaning that the final layer tokens, to which WiCLP is applied, already incorporate context from previous tokens. As a result, a token within a given window also carries information about many tokens outside that window.
>
> While we agree that increasing the window length can enhance performance, our experiments show that even smaller window lengths achieve high compositional accuracy. For instance, the CLP method, which uses a window length of 1 (only focusing on a single token), still demonstrates strong results, underscoring the effectiveness of our approach.
>
> > There are no failure case visualizations ...
>
> The reviewer has raised an excellent point, and we greatly appreciate their constructive suggestion. We will add a dedicated section to the paper discussing the failure cases of our model, along with an in-depth analysis of these instances. Additionally, we will incorporate an evaluation of the specific categories where our method performs well and those where it falls short.
>
> > Whether there are examples in the tested datasets that the proposed method failed to generate the expected compositions?
>
> We acknowledge that not all images generated using WiCLP are flawless, and we will include examples of failure cases in the final version of the paper. However, it is important to emphasize that no method or model consistently produces "fully correct" compositional images across an entire dataset, as reflected in their relatively low VQA scores. Despite these challenges, our method can easily be integrated with any model to significantly enhance their compositional performance.

---

> > ### Comment · Reviewer_tRxw · 2024-11-26
> >
> > Thank you for the responses. While reading other reviewers' responses, I played with the gradio link the authors sent to reviewer sPe7, I found that for most of the prompts, SDXL+WiCLIP shows better attribute binding of color, shape comparing to SDXL, but often generate the wrong number of objects, i.e., the prompts are all in the form of "one A one B", but SDXL+WiCLIP usually generates many A and B. This seems contradicts with the counting performance comparison shown with SD v 1.4. So my concern is still that the proposed linear layer finetuning method, with a simple locality constraint, may not be fundamentally addressing the token attention assignment problem, since it is unclear how the proposed method helps the underlying representation parsing of the sentence. Therefore I will maintain my current rating.

---

> > > ### Author Response · Authors · 2024-11-27
> > >
> > > Thank you for taking the time to explore our demo and share your thoughtful feedback. We genuinely appreciate your acknowledgment of the model's improved attribute binding accuracy. You've highlighted a valid concern regarding its current limitations, particularly the tendency to generate multiple objects. To clarify, the model has been trained solely on the color category of the T2I-CompBench dataset. This focused training explains its strong performance in color attribute binding while revealing weaknesses in numerical compositionality. Expanding the training to encompass a broader range of attributes and categories is crucial for achieving more balanced performance. We plan to address this in future work and will include comprehensive results in the final version of the paper.
> > >
> > > Thank you once again for your valuable insights. We hope our responses have addressed your concerns and provided clarity, potentially influencing your score or rating in a positive direction.

---

### Official Review · Reviewer_PsCB · 2024-11-02

**Soundness:** 2
**Presentation:** 1
**Contribution:** 1
**Rating:** 3
**Confidence:** 5

**Summary:**

This paper investigates compositionality-based failure modes in text-to-image generative models, identifying imperfect text conditioning with the CLIP text encoder as a key factor hindering the generation of high-fidelity compositional scenes. To address this, the authors propose a simple yet effective baseline, WiCLP, which fine-tunes a linear projection within CLIP's representation space. WiCLP achieves superior performance on compositional image generation benchmarks while maintaining a low FID score across a wide range of clean prompts.

**Strengths:**

1. The paper provides a detailed analysis of CLIP text embeddings, examining the response and alignment between tokens, which accurately pinpoints the root of compositionality issues in SD 1.5 and SD v2.
2. The experimental results convincingly demonstrate the effectiveness of the proposed improvement, providing clear evidence of its advantages over existing methods.

**Weaknesses:**

1. My primary concern is the limited innovation. The proposed solution merely involves a projection fine-tuning of CLIP text embeddings, with WiCLP simply aggregating information from neighboring tokens. Ultimately, the method only fine-tunes text embeddings on a selected dataset using Stable Diffusion, lacking sufficient technical novelty.

2. The paper includes limited comparisons. There are no qualitative results of other baselines such as SDXL, nor comparisons with recent compositional generation methods such as RPG[1], Realcompo[2]. The baselines provided seem somewhat outdated, and the absence of recent methods makes it challenging to gauge the true impact of the proposed approach.

3. The experiments are limited. By fixing the U-Net and text encoder and only fine-tuning the text embeddings, the paper restricts itself, as fine-tuning any component on the selected data might yield compositional improvements. A comparison with alternative fine-tuning approaches would help validate whether adjusting the text embeddings specifically yields the best results, rather than gains simply from better-selected samples.

4. Limited impact. The baseline used is Stable Diffusion v2, which is outdated, and the approach centers around fine-tuning the CLIP text encoder. However, most modern generative models, like SD3 and Flux, now use T5-based encoders. Thus, it is uncertain if the proposed method would have practical utility in today’s models.

5. Poor presentation. The paper has significant issues in its writing: quotation marks are inconsistently formatted, Figure 5 lacks labels on the left, making it hard to interpret, and Figure 7 is very low resolution.

[1]Yang, Ling, et al. "Mastering text-to-image diffusion: Recaptioning, planning, and generating with multimodal llms." Forty-first International Conference on Machine Learning. 2024.

[2]Zhang, Xinchen, et al. "Realcompo: Dynamic equilibrium between realism and compositionality improves text-to-image diffusion models." arXiv preprint arXiv:2402.12908 (2024).

**Questions:**

1. I am curious whether fine-tuning other parts of Stable Diffusion could achieve similar improvements. It would be beneficial to evaluate which component adjustments yield the most effective or efficient results.

2. How would the proposed method be adapted for T5-based diffusion models, such as Pixart-Alpha? Is it likely to produce comparable results in these architectures?

---

> ### Author Response · Authors · 2024-11-22
>
> We appreciate the reviewer’s thoughtful and constructive comments. Below, we address each of their comments individually:
>
> > My primary concern is the limited innovation. The proposed ...
>
> We note that this simple and lightweight fine-tuning approach, inspired by our observations of the CLIP text encoder's shortcomings, results in significant improvements. Its lightweight nature allows for both data-efficient and parameter-efficient fine-tuning, making it particularly well-suited for compositionality tasks. During the rebuttal, we conducted an ablation study across various fine-tuning parameter sets and found that fine-tuning a linear layer is not only highly efficient in terms of parameters and computational speed but also achieves comparable or even superior improvements. compared to other fine-tuning methods that modify significantly more parameters. These other methods often lead to a noticeable degradation in model performance on metrics such as FID score, as discussed in our paper.
>
> The tables below present the evaluation results for training various parameter sets:
> | Fine-tuning Parameters | VQA Score [Color Category] |
> |:-:|:-:|
> |Baseline (SD v1.4) | 0.3765 |
> | full CLIP | 0.5173 |
> | Last Layers of CLIP + WiCLP | 0.5497 |
> | WiCLP | 0.5383 |
>
> | Fine-tuning Parameters | VQA Score [Color Category] |
> |:-:|:-:|
> | Baseline (SD v2) | 0.5065 |
> | UNet (GORS from T2I-CompBench) | 0.6414 |
> | WiCLP | 0.6500 |
>
> > The paper includes limited comparisons. There are no qualitative results of other baselines such as SDXL …
>
> Thank you for your suggestion. We agree additional baseline models would strengthen our paper, and as such, have conducted new experiments on SDXL. The results demonstrate substantial improvements in compositionality scores. Notably, the performance of SD-XL, enhanced by our method, was comparable to that of the state-of-the-art SD-v3, showcasing significant progress. Below, we present the results of applying WiCLP to SD-XL:
>
> | | Color | Texture | Shape |
> |:-:|:-:|-|-|
> | Baseline SD-XL | 0.5770 | 0.5217 | 0.4666 |
> | SD-XL + WiCLP | **0.7801** | **0.6557** | **0.5166** |
>
> Due to limited time, we were only able to run experiments on SD-XL. However, we plan to conduct similar experiments on SD-v3 and anticipate observing comparable improvements. These results will be included in the final version of the paper.
>
> We also thank the reviewer for highlighting other recent compositional generation methods, such as RPG and RealCompo. We will ensure these baselines are included in our comparison table in the final version of the paper.
>
> > The experiments are limited. By fixing the U-Net and text encoder ...
>
> We thank the reviewer for raising this point. We note that our focus on only finetuning the text embeddings is motivated directly by the significant issues we identify in Sections 3 and 4. Nonetheless, during the rebuttal, as discussed in the above section, we conducted new experiments finetuning broader sets of parameters. In order to achieve a very good trade-off between efficiency and performance, we found that our WiCLP achieves the best trade-off.
>
> > Limited impact. The baseline used is Stable Diffusion v2, which is outdated, and the approach centers around fine-tuning the CLIP text encoder ...
>
> We appreciate the reviewer’s comment. While our paper primarily focuses on the CLIP text encoder, which is commonly used in many recent text-to-image models, our method is also applicable to T5-based T2I models. During the rebuttal, we conducted experiments on DeepFloyd-I-M, which utilizes a T5-XXL text encoder and observed similar issues in compositionality.
>
> For instance, with the prompt “a red book and a yellow vase”, the baseline model achieved a VQA score of 0.4350. By optimizing the text embedding space, based on the method discussed in Section 4, we significantly improved this score to 0.9121, demonstrating that the text embedding space of this model can be further refined. Moreover, applying WiCLP to DeepFloyd-I-M improved its VQA performance across all 300 evaluation prompts, increasing the score from 0.4636 to 0.5155. These results highlight the broader applicability and effectiveness of our method.
>
> Furthermore, we conducted experiments on newer models like SD-XL, achieving significant improvements in compositionality, as discussed previously.
>
> > The paper has significant issues ...
>
> We sincerely thank the reviewer for their valuable feedback and apologize for any issues in the presentation. We will make every effort to address these concerns and improve the overall clarity and quality of the presentation in the final draft.
>
> > I am curious whether fine-tuning other parts of SD could achieve similar improvements…
>
> Please refer to the sections above where we discuss new experiments on fine-tuning across different parameter sets.
>
> > How would the proposed method be adapted for T5-based …
>
> Please refer to the sections above where we discussed new experiments on applying WiCLP to DeepFloyd.

---

> > ### Author Response · Authors · 2024-11-27
> >
> > Dear Reviewer PsCB,
> >
> > Thank you for your thoughtful and constructive feedback on our paper. With only a few days remaining in the discussion period, we kindly request that you review our responses to ensure we have adequately addressed your concerns. If our responses meet your expectations, we would be deeply grateful if you could reconsider your rating or score.
> >
> > Your engagement and constructive input have been invaluable, and we truly appreciate your time and effort in supporting this process.

---

### Official Review · Reviewer_WVTN · 2024-11-02

**Soundness:** 3
**Presentation:** 3
**Contribution:** 2
**Rating:** 5
**Confidence:** 4

**Summary:**

This paper aims to mitigate the compositional issues in recent CLIP-based diffusion models, such as Stable Diffusion. Specifically, the authors first show that the output space of the CLIP text encoder is sub-optimal. Then, they propose to fix it by using a window-based linear projection for text embeddings from the CLIP model. Experiments have demonstrated the effectiveness of their approach.

**Strengths:**

1. The idea is easy to understand. The CLIP model may not capture some key concepts or attributes in a prompt. Thus, applying a window-based linear projection could help reweight the text embeddings, which improves prompt alignment in text-to-image generation.

2. The proposed window-based linear projection brings a clear improvement in terms of color, texture, and shape, as demonstrated in Table 1.

3. This paper is easy to follow. Figures have well demonstrated their core idea.

**Weaknesses:**

1. There exist some training-free approaches to reweight the text embedding for better prompt alignment, such as enhancing the attention on a few words, e.g., (tuxedo) or (tuxedo:1.21) in stable-diffusion-webui, where the number indicates the strength of word embedding. This approach reweights the text embeddings from CLIP encoder, in a way that it slightly changes the overall text condition signals for the subsequent diffusion process. Compared to it, the introduced WiCLP in this paper trains an additional layer which achieves something similar but may specifically target one model only. As the aforementioned approach has been widely adopted in the text-to-image generation community, the authors are encouraged to add a brief discussion with it [A].

2. Practical concerns. This approach requires training an additional linear projection, which could be impossible if we only have the released model weights and cannot access the pretraining dataset. According to Section D.1, the training cost is also not negligible (25K steps on one RTX A5000 GPU). It would be good to know if the trained linear projection can be generalized to other diffusion models as well, such as those finetuned/stylized diffusion model [B,C].

[A] https://github.com/AUTOMATIC1111/stable-diffusion-webui/wiki/features#attentionemphasis

[B] https://huggingface.co/nitrosocke/Ghibli-Diffusion

[C] https://huggingface.co/playgroundai/playground-v2.5-1024px-aesthetic

**Questions:**

Please see the weakness.

---

> ### Author Response · Authors · 2024-11-22
>
> We appreciate Reviewer WVTN for their time and valuable feedback. We will address each of their comments individually below:
>
> >There exist some training-free approaches to reweight the text embedding …
>
> We thank the reviewer for highlighting this point. We will ensure a discussion of these methods is included in the final version of the paper. Additionally, as part of the rebuttal, we conducted new experiments with the Prompt-to-Prompt (P2P) [1] method, which adjusts the attention mechanism by reweighting it. Specifically, we used P2P to increase the model's attention to all objects and attributes in the input prompt using a weighting approach. The table below presents the results, comparing P2P and WiCLP on SD v1.4.
>
> |           |   P2P  | WiCLP  |
> |:---------:|:------:|--------|
> | VQA Score | 0.3493 | **0.5850** |
>
> As the results show, our method significantly outperforms P2P. Moreover, it is worth noting that our method is model-agnostic and can be applied to any model, not just a specific one.
>
>
> > Practical concerns. This approach requires training an additional linear projection …
>
> We should note that in our framework, access to the pretraining dataset is not required; the projection is trained using prompt/image pairs collected via open-source generators.
>
> We understand the reviewer’s concern regarding the applicability of our method to personalized and fine-tuned models. Personalized models, often fine-tuned with LoRA, adjust the key and value matrices of cross-attention layers to enable the UNet to generate personalized images. Directly using a pre-trained WiCLP projection in such models does not work, as WiCLP modifies text embeddings to better handle compositional prompts, which may conflict with the personalized key/value projections. However, we can achieve this goal by retraining the WiCLP for the fine-tuned and personalized models. To this end, we will release the code and data needed for retraining.
>
> Additionally, our method is computationally efficient. With optimized training techniques, a projection layer for a model as large as SD-XL can be trained on a single A6000 GPU in under six hours. Note that WiCLP is model-agnostic and not restricted to original pre-trained models. In fact, it could be applied on top of personalized models.
>
> [1] Hertz, Amir, et al. "Prompt-to-prompt image editing with cross attention control." arXiv preprint arXiv:2208.01626 (2022).

---

> > ### Comment · Reviewer_WVTN · 2024-11-25
> > **Official Comment from Reviewer**
> >
> > Thanks for the authors' rebuttal. I have carefully read the comments from other reviewers. Overall, under the specific research question, the work has demonstrated good effectiveness, as shown in the author's rebuttal and their experiments. However, I think a common concern shared by the other reviewers is the practical value of this work, as it may not be very general to well-trained diffusion models or more convenient if comparing to training-free approaches. Considering the above, I would like to maintain my initial rating.

---

> > > ### Author Response · Authors · 2024-11-25
> > >
> > > We sincerely thank the reviewer for their valuable feedback and for recognizing the effectiveness of our approach. During the rebuttal period, we conducted additional experiments on a more recent and well-trained model, SD-XL, and observed significant improvements. Specifically, WiCLP achieved substantial enhancements in compositionality scores, with increases of 13% and 20% in the texture and color categories, respectively. Remarkably, incorporating WiCLP into SD-XL results in performance comparable to the SOTA SD-v3 model, reflecting notable progress.
> > >
> > > Below, we present the results of applying WiCLP to SD-XL:
> > > |                |  Color | Texture | Shape  |
> > > |:--------------:|:------:|---------|--------|
> > > | Baseline SD-XL | 0.5770 | 0.5217  | 0.4666 |
> > > |  SD-XL + WiCLP | **0.7801** | **0.6557**  | **0.5166** |
> > >
> > > Due to time constraints, we focused our experiments on SD-XL. However, we plan to extend these experiments to SD-v3 and anticipate observing comparable improvements, which will be included in the final version of the paper.
> > >
> > > Furthermore, as discussed above, our results demonstrate that WiCLP significantly outperforms training-free methods, achieving over 23% higher VQA scores.
> > >
> > > We hope these new findings address the reviewer’s concerns regarding the practical value and generalizability of our method. We would greatly appreciate it if the reviewer could reconsider their evaluation or let us know if any additional concerns remain.

---

### Official Review · Reviewer_sPe7 · 2024-11-04

**Soundness:** 2
**Presentation:** 2
**Contribution:** 1
**Rating:** 5
**Confidence:** 3

**Summary:**

This paper investigates why early versions of stable diffusion (v1.4 and v2) are bad at composition. The authors found two sources: 1) CLIP text embeddings have wrong attentions between words, 2) the CLIP text embeddings fail to fully drive the UNet to generate compositional images. The authors propose two patches for the two sources, respectively: 1) zero-shot attention reweighting, 2) adding a linear projection layer to the CLIP text embeddings to make it drive the UNet more accurately.

**Strengths:**

1) The proposed methods are simple and easy to implement.
2) The proposed method are validated to be effective in experiments.

**Weaknesses:**

1) Training a linear projection layer on top of CLIP text embedding seems ad-hoc. Especially, the authors focused on a few types of attributes:  color, texture, shape. However, in practice, we cannot limit compositional prompts to these few types. If we want the projection layer to handle more types, we need to train the layer on more diverse instances, which demand increasing manual efforts.

2) The newer models, e.g., SDXL, SD 3.5, and FLUX, have been much better at composition. This indicates that the source of bad composition is the SD1.4/2.0 models were not well designed or well trained. Therefore, systematic solutions like redesigning and retraining the model should be our main pursuit for solving compositionality. **Patching a bad model** could work as a temporary solution to some degree, but I'm not sure if this direction is worth investing significant amount of time and efforts.

**Questions:**

N/A

---

> ### Author Response · Authors · 2024-11-22
>
> We appreciate Reviewer sPe7 for their time and valuable feedback. We will address each of their comments individually below:
>
> > Training a linear projection layer on top of CLIP text embedding seems ad-hoc. Especially, the authors focused on a few types of attributes…
>
> We note that attribute binding, which our work focuses on, is a core aspect of compositionality, encompassing many subcategories like colors, textures, and shapes. While we observe the approach to be very effective for these areas, we appreciate the reviewer’s suggestion to explore how broadly our method can be applied. To this end, we conduct a new set of experiments during the rebuttal to see if our lightweight method can prove effective in numeracy (i.e. counting) – a markedly distinct aspect of compositionality. The table below presents the results of our method on the Numeracy split of T2I-CompBench.
>
> Through the rebuttal, we evaluated our method on the Numeracy category of T2I-CompBench, and the results are presented below.
> |                    | Disentangled BLIP-VQA Score |
> |:------------------:|:---------------------------:|
> | Baseline (SD v1.4) |            0.4413           |
> |   SD v1.4 + WiCLP  |            **0.4890**           |
>
> Indeed, we find that WiCLP again provides notable gains, showcasing the generalizability of our method over different compositional challenges.
>
> > The newer models, e.g., SDXL, SD 3.5, and FLUX, have been much better at composition. This indicates that the source of bad composition …
>
> We thank the reviewer for highlighting this point. We completely agree that newer models, such as SD-XL, generally achieve better compositionality. However, our method can further enhance their performance. To demonstrate this, we conducted experiments on SD-XL. The results show substantial improvements in compositionality scores, including large leaps of 13% and 20% for the texture and color categories. In fact, adding WiCLP to SD-XL leads to performance comparable to the SOTA SD-v3, reflecting significant progress. While the mitigations the reviewer mentions improve compositionality, our new results suggest the WiCLP may offer complementary gains.
>
> We will include the results for these newer models in the final version of the paper. Below, we present the results of applying WiCLP to SD-XL:
>
> |                |  Color | Texture | Shape  |
> |:--------------:|:------:|---------|--------|
> | Baseline SD-XL | 0.5770 | 0.5217  | 0.4666 |
> |  SD-XL + WiCLP | **0.7801** | **0.6557**  | **0.5166** |
>
> Due to limited time, we were only able to run experiments on SD-XL. However, we plan to conduct similar experiments on SD-v3 and anticipate observing comparable improvements. These results will be included in the final version of the paper.
> > … redesigning and retraining the model should be our main pursuit for solving compositionality …
>
> We fully agree with the reviewer that redesigning and retraining models is crucial for improving compositionality, as evidenced by the trend of newer models achieving better performance. However, even the most recent state-of-the-art models still struggle in many compositional scenarios. This highlights the need for complementary methods, such as ours, to further address and mitigate these persistent issues. Our new experiments on SD-XL demonstrate that our method is not limited to enhancing weaker models; rather, WiCLP can also be effectively integrated into newer, more advanced models, leading to significant improvements in their performance.

---

> > ### Comment · Reviewer_sPe7 · 2024-11-25
> > **Thanks for the response**
> >
> > The extra experimental results look impressive. I'd suggest the authors make an anonymous huggingface demo to compare the results "before" and "after" applying the CLIP patch. I'd be happy to raise my rating if the improvement is indeed obvious. Thanks. Before that, I will take these results with a grain of salt.

---

> > > ### Author Response · Authors · 2024-11-25
> > >
> > > We sincerely thank the reviewer for their valuable feedback and are glad that you found our new results compelling. To address your suggestion, we have created an anonymous Gradio demo showcasing qualitative examples of our method, both "before" and "after" applying WiCLP on top of SDXL. Please note that the provided link will remain active for the next 48 hours.
> > >
> > > Link: [Gradio Demo](https://fee25fc35b16c92796.gradio.live/)

---

> > > > ### Comment · Reviewer_sPe7 · 2024-11-26
> > > > **Appreciate the demo**
> > > >
> > > > I appreciate the authors paid effort to set up the demo. I've tried a few examples and indeed the attribute binding accuracy improves a lot. However, just as reviewer tRxw points out, now SDXL tends to generate multiple objects of the same type. One more issue I noticed is, the generated objects tend to be simplistic and toyish, and the background is often blank, while the original model generates more complicated and realistic objects with complex background. (Examples are "a red book and a yellow vase", and "a brown bird and a blue bear"). It seems that during fixing the attribute binding, the projection layer discards a lot of semantic richfulness. I'd regard this work as an intermediate step towards a useful module. At its current form, I'd keep my current rating. Thanks.

---

> > > > > ### Author Response · Authors · 2024-11-27
> > > > >
> > > > > Thank you for taking the time to explore our demo and for providing thoughtful feedback. We truly appreciate your recognition of the model’s improved attribute binding accuracy. You’ve raised an important point regarding its current limitations, particularly the generation of multiple objects. We’d like to clarify that the model in its current form has been trained exclusively on the color category of the T2I-CompBench dataset. This focus is why it performs exceptionally well on color attribute binding but struggles with numerical compositionality. To achieve balanced performance across all categories, additional training on a broader set of attributes and categories is essential. We plan to address this in future work and include comprehensive results in the final version of the paper.
> > > > >
> > > > > Thank you again for your insightful comments. I hope our responses have effectively addressed your concerns and provided clarity, potentially influencing your score or rating positively.

---

### Note · Authors · 2024-12-10

I have read and agree with the venue's withdrawal policy on behalf of myself and my co-authors.